# Metabolic Potential of Some Functional Groups of Bacteria in Aquatic Urban Systems

**Bianca Ojovan [1], Rodica Catana [1], Simona Neagu [1], Roxana Cojoc [1], Anca Ioana Lucaci [1,*], Luminita Marutescu [2,3], Larisa Florescu [1], Robert Ruginescu [1], Madalin Enache [1] and Mirela Moldoveanu [1]**

[1] Institute of Biology Bucharest of Romanian Academy, 296 Splaiul Independentei, 060031 Bucharest, Romania; bianca.ojovan@ibiol.ro (B.O.); rodica.blindu@ibiol.ro (R.C.); simona.neagu@ibiol.ro (S.N.); roxana.cojoc@ibiol.ro (R.C.); larisa.florescu@ibiol.ro (L.F.); robert.ruginescu@ibiol.ro (R.R.); madalin.enache@ibiol.ro (M.E.); mirela.moldoveanu@ibiol.ro (M.M.)

[2] Department of Botany and Microbiology, Faculty of Biology, University of Bucharest, 1-3 Aleea Portocalelor Str., 60101 Bucharest, Romania; lumi.marutescu@gmail.com

[3] Research Institute of University of Bucharest—ICUB, 91-95 SplaiulIndependentei, 010271 Bucharest, Romania

* Correspondence: ioanalucaci12@yahoo.com

**Abstract:** This study analyzed the metabolic potential of some functional groups of bacteria in aquatic urban systems and evaluated the abundance of communities of total heterotrophic bacteria in the water in relation to the monitored physico-chemical factors. The results obtained showed seasonal differences, especially in spring. The high values of the abundance of heterotrophs in winter are related to human activity at the sampling stations. Screening for four types of extracellular hydrolytic enzyme with potential for degradation of organic matter (amylases, lipases, proteases and cellulases) led to the conclusion that lipolytic bacteria were dominant in the studied ecosystems, while proteolytic bacteria were observed in low numbers, but were present in urbanized areas. The presence of cellulolytic bacteria is correlated with the development of macrophytic vegetation. The aim of the present study was oriented towards the evaluation of the anthropogenic input in several lakes surrounding Bucharest in the Nord-Eastern region. These urban ecosystems are generated as a requirement of city development. The microbiological and general enzymatic approaches generated some novel results concerning the pollution degree of aquatic urban ecosystems and could be considered as a platform for further investigation.

**Keywords:** urban ecosystems; heterotrophic bacteria; extracellular hydrolases; metabolic potential; Colentina River

## 1. Introduction

In aquatic ecosystems, communities of microorganisms are essentially involved in the process of organic matter decomposition. The importance of microorganisms in the organic matter pathways along the food web has been reconsidered beginning with the recognition of the existence of the microbial loop by Azam [1]. Although microorganisms in nature have been studied for centuries, in recent decades they have been recognized as drivers of nutrients and energy, especially in the oceans. Since the 20th century, several important publications have stated the role of microorganisms in the primary production and consumption of dissolved organic substances [2–4]. These findings supported the hypothesis that microorganisms provide a link in the food web between phytoplankton, dissolved nutrients, and zooplankton. Several studies had as their main purpose the investigation of the processes by which microorganisms decompose organic matter and restore to other trophic levels the nutrients necessary for the functioning of the ecosystem. At the same time, the role of microorganisms in the natural purification of waters, clearing of lakes, and removal of pollutants was investigated [5].

Among microorganisms, heterotrophic bacteria represent the main component responsible for the decomposition and conversion of macromolecular detritus which provides

the nutrients needed to higher trophic levels in aquatic ecosystems [6]. Most substrates found in the environment require cleavage into monomers or small oligomers in order to be assimilated and processed into cells [7]. Cleavage is generally performed by hydrolytic enzymes extracellularly secreted by microorganisms. Screening of extracellular enzymes produced by bacteria was performed mainly from extreme natural environments, except for a few works in which the distribution of microorganisms producing extracellular enzymes in soil, water or plants was analyzed [7,8]. The main aim of these studies was to determine the microbial abundance, isolation and characterization of microorganisms for biotechnological applications [9,10]. Bacterial isolates, having amylolytic, lipolytic and cellulolytic activity were obtained from thermal springs, with temperature values of 55–70 °C [11]. By reporting the abundance distribution of certain types of bacterial enzymes, it was possible to appreciate specific functional profiles of the microbial communities, which varied with the temperature and available substrates, or even under the influence of anthropogenic impact [9,12]. Although there are more important papers onthe enzymatic equipment of microorganisms such as proteases, lipases, amylases, the expression of these enzymes hasnot been completely studied and understood [13]. Also, cellulosolytic enzymes have a major contribution in aquatic ecosystems where a rich community of higher plants is developed. The reeds present on the lake shores degradingat the end of life cycle enrich the water mass with plant residues, containing cellulose.

Generally, urban ecosystems are novel research fields because their ecological functioning is not fully known. Most urban aquatic ecosystems are under the impact of human communities. With the status in different degrees of anthropization, controlled, arranged or even degraded, urban rivers and lakes are constantly changing their ability to provide ecological services to human communities. The organic matter accumulated in these types of ecosystem, combined with the presence of various pollutants, exceeds the buffering capacity of ecosystems. The role of heterotrophs in degrading organic matter into simple inorganic compounds within biogeochemical cycles is related to the ability of these microorganisms to produce extracellular hydrolytic enzymes. Depending on the compounds available in the environment, bacteria produce several categories of enzymes. In general, bacteria synthesize constituent enzymes, necessary to use a simple source of organic carbon such as glucose. However, they have the ability to produce enzymes necessary for the processing of other substrates available in the environment, if the concentration of the preferred falls below the critical threshold [12].

The present study aimed to evaluate the production capacity of microbial extracellular hydrolytic enzymes potentially involved in the organic matter degradation processes in the lakes along Colentina River. This investigation had two specific objectives, the first addressed the abundance of total heterotrophic bacteria in water in relation to the physico-chemical factors; the second one focused on the screening of extracellular hydrolytic enzymes with the degradation potential of organic matter (amylases, lipases, proteases and cellulases).

## 2. Materials and Methods

### 2.1. Description of Sampling Site

The Colentina River (101 km) crosses Ilfov County and Bucharest in the form of a chain of 15landscaped lakes connected by a system of hydrological control locks.Tenstations were established: a station (SP0) in the unchanged area of the river considered the reference area, two stations/lake in Mogoșoaia (SP2 and SP3), Plumbuita (SP4 and SP5) and Fundeni (SP6 and SP7) lakes, a station in Cernica lake (SP8) and a station near to Cernica lock system (SP9). Also, a sampling station (SP1) was established on the Crevediabranch (tributary of Colentina) [14,15]. Thus, the result was a spatial sampling program in the direction of the rural–urban gradient, up to the downstream area of the Colentina River.

A large part of the shores of these lakes are adjacent to households and some are included in parks [16]. For this reason, the Colentina river has acquired over time an important role as agreement area. Being accessible to human population, the ecosystem

services provided by the river (agreement, fish exploitation, urban temperature regulation) have a great importance.

### 2.2. Sampling

Four sampling campaigns were carried out in March, July, September and November to cover the seasonal periods from the 10sampling stations that were set up along the Colentina River.

Physico-chemical parameters were measured in situ. Temperature, salinity, conductivity, oxidation-reduction potential (ORP), pH, total dissolved solids (TDS) and dissolved oxygen (DO) were measured with a Hanna Instruments HI 9828 multi-parameter, while the depth and transparency of the lakes were determined using the Secchi disk. Also, the water velocity was determined using a basic flowmeter, Geopacks Flowmeter (MFP51).

For microbiological analysis, a total of 40 samples (one sample/sampling station/season) of approximately 200 mL of water each were taken in sterile glass containers. The samples were taken from the water column, stored at 4 °C during field measurements, then transported to the laboratory and processed in aseptic conditions within a maximum of 24 h.

### 2.3. Media and Cultivation

The pour plate method was used both to determine the number of heterotrophic microorganisms and to highlight the production of extracellular hydrolytic enzymes [17]. Incubation was performed at 37 °C for at least 48 h with the possibility of extending the duration for recording the production of extracellular enzymes.

Four enzymatically different types of culture media were selected for enzymatic screening containing specific substrates [18,19]. Thus, agarized media with 1% starch, 1% casein, 1% Tween80 and 0.5% carboxymethylcellulose, respectively, were used in order to select bacteria able to produce amylases, proteases, lipases and cellulases. The colonies present on each plate were counted, the results being expressed in colony forming units (CFU) per 1 mL sample.

### 2.4. Data Analysis

Statistical processing was performed using XLStat (2013) Add-in software. Descriptive statistical analysis and blox-plots, the ANOVAtest and RDA (redundancy analysis)were performed. From the descriptive statistics, minimum, maximum, average, Variance and Standard deviation were selected [20]. Box-plot graphs also have the role of visually highlighting the statistical characteristics of the data set in a sample [21]. Based on the one-way ANOVA test, the significance of spatial and temporal variations in heterotrophic bacterial abundance was assessed [22]. The influence of physico-chemical parameters on the distribution of microbial communities was assessed with a multivariate analysis [23,24]. For this purpose, the variables were transformed by logarithm, according to the log relation ($n$ + 1), to obtain a normal distribution. Also, multicollinearity matrix was generated. After statistically testing the gradient length, the use of the linear RDA ordering method, suitable for short gradients, was established. For this analysis we used two matrices: a matrix composed of response variables (density of heterotrophic bacteria, amylolytic, proteolytic, lipolytic and cellulolytic bacteria) and a matrix composed of explanatory variables (independent) represented by physico-chemical factors (conductivity, ORP, temperature, turbidity, depth and pH). The significance of the ordering axes was tested using the Monte-Carlo permutation test with 999 permutations. The test indicated $p < 0.05$.

## 3. Results

### 3.1. Environmental Parameters

The depth of the selected lakes, being controlled by the lock system of the Colentina River, varied in between 0.20–1.80 m. This water flow control system also influenced water velocity (m/s) which showed low values (Table 1). Indirectly, transparency was

also influenced by this system, as a factor positively correlated with depth ($R^2 = 0.493$; $p = 0.001$). Regarding the spatial and temporal characteristics of the parameters, the factors that varied significantly ($p < 0.05$) among ecosystems (depth, water velocity and DO) and the factors that varied seasonally (temperature, conductivity, salinity, TDS, ORP, pH and turbidity) were highlighted.

**Table 1.** Summary statistics of physico-chemical parameters of Colentina river lakes chain.

| Variable | Minimum | Maximum | Mean | Std. Deviation |
| --- | --- | --- | --- | --- |
| Depth (m) | 0.20 | 1.80 | 0.81 | 0.42 |
| Water velocity (m s$^{-1}$) | 0.05 | 0.19 | 0.07 | 0.03 |
| Temperature (°C) | 7.42 | 29.60 | 17.37 | 6.41 |
| Transparency (m) | 0.20 | 1.00 | 0.48 | 0.18 |
| Turbidity (UM) | 8.00 | 94.60 | 19.65 | 13.59 |
| Conductivity (mS cm$^{-1}$) | 387.00 | 710.00 | 510.23 | 86.78 |
| Salinity (UM) | 0.19 | 0.35 | 0.25 | 0.04 |
| Total dissolved solids | 194.00 | 355.00 | 255.00 | 43.44 |
| DO (mg $O_2 L^{-1}$)-surface | 2.75 | 22.16 | 8.71 | 4.83 |
| DO (mg $O_2 L^{-1}$)-bottom | 2.72 | 22.33 | 8.03 | 4.94 |
| DO (%) saturation-surface | 25.00 | 211.70 | 91.52 | 49.58 |
| DO (%) saturation-bottom | 24.00 | 212.50 | 82.85 | 47.94 |
| pH | 6.63 | 10.07 | 8.63 | 0.72 |
| ORP | −168.60 | 11.20 | −95.79 | 37.26 |

### 3.2. Heterotrophic Bacteria

In the studied ecosystems, the abundance of heterotrophic bacteria varied between 25 CFU mL$^{-1}$ (Cernica, spring) and $1.2 \times 10^6$ CFU mL$^{-1}$ (Lake Mogoșoaia, autumn) (Table 2). This high value was unusual because it was higher than the values quantified throughout the study period and can be considered an outlier from a statistical point of view. Also, the second value in terms of magnitude order is recorded in Lake Mogoșoaia ($2.3 \times 10^5$ CFU mL$^{-1}$). The values recorded in the other lakes do not reach the values from Lake Mogoșoaia in any of the seasons. There is an increased value in spring ($5.6 \times 10^3$ CFU mL$^{-1}$) in station SP0, located on the Colentina River, in the natural area of the river. It is the only high value compared to other lakes in the spring season.

**Table 2.** Heterotrophic bacterial abundance (CFU mL$^{-1}$).

| | | Spring | Summer | Autumn | Winter |
| --- | --- | --- | --- | --- | --- |
| Colentina River | SP0 | $5.6 \times 10^3$ | $1.7 \times 10^3$ | $1.3 \times 10^3$ | $2.3 \times 10^3$ |
| Crevedia branch | SP1 | $3 \times 10^2$ | $3.6 \times 10^3$ | $4.6 \times 10^3$ | $1.6 \times 10^3$ |
| Mogoșoaia | SP2 | $3.5 \times 10^2$ | $1.1 \times 10^3$ | $1.2 \times 10^6$ | $1.5 \times 10^3$ |
| | SP3 | 76.5 | $4.3 \times 10^3$ | $2.3 \times 10^5$ | $1.6 \times 10^3$ |
| Plumbuita | SP4 | $5 \times 10^2$ | $3 \times 10^3$ | $1.6 \times 10^3$ | $1.7 \times 10^4$ |
| | SP5 | 36 | $3.4 \times 10^3$ | $1.3 \times 10^3$ | $1.2 \times 10^4$ |
| Fundeni | SP6 | 73.5 | $3.5 \times 10^3$ | $2.3 \times 10^3$ | $2.4 \times 10^3$ |
| | SP7 | 75 | $5.9 \times 10^2$ | $6 \times 10^3$ | $7.7 \times 10^3$ |
| Cernica | SP8 | 25 | $1.1 \times 10^3$ | $3.3 \times 10^3$ | $2.2 \times 10^3$ |
| | SP9 | $10.3 \times 10^2$ | $10.3 \times 10^3$ | $7.5 \times 10^3$ | $1.5 \times 10^2$ |

A special situation was recorded in the stations on Plumbuita Lake, when the highest values of the year ($1.7 \times 10^4$, respectively $1.2 \times 10^4$ CFU mL$^{-1}$) were registered in the winter season (Table 2).

Based on the one-way ANOVA test, the significance of spatial and temporal variations in heterotrophic bacterial abundance was assessed. The results showed only significant seasonal differences ($F_{(3,36)} = 9.11$; $p = 0.000$). The post hoc Tukey test (HSD) showed that spring was the season in which the abundance of bacteria differed from the other seasons (Table 3). From Table 2 it can be seen that the changes of the environmental parameters

from spring to summer caused increases of more than 40 times of the abundances of heterotrophs at stations SP3, SP5, SP6 and SP8. The transition from spring to summer was the period of intense development of heterotrophs in all stations. This development continued with lower intensity in the next two seasons under the influence of local environmental conditions. The significant difference between winter and spring was determined by a higher abundance of heterotrophs in the winter season, especially in ecosystems with urban influences (stations SP1- SP8). On the other hand, stations SP0 and SP9 showed moderate changes in abundance from one season to another, both being characterized by conditions similar to those recorded in natural environments (Table 2).

**Table 3.** Post hoc Tukey (honestly significant difference, HSD) test on seasonal development of heterotrophic bacteria.

| Contrast | Difference | Standardized Difference | Critical Value | Pr > Diff | Significant |
|---|---|---|---|---|---|
| winter vs. spring | 1.1027 | 3.4833 | 2.6933 | 0.0069 | Yes |
| winter vs. summer | 0.0238 | 0.0751 | 2.6933 | 0.9998 | No |
| autumn vs. spring | 1.6036 | 5.0658 | 2.6933 | <0.0001 | Yes |
| autumn vs. summer | 0.5247 | 1.6576 | 2.6933 | 0.3605 | No |
| autumn vs. winter | 0.5010 | 1.5825 | 2.6933 | 0.4011 | No |
| summer vs. spring | 1.0789 | 3.4082 | 2.6933 | 0.0085 | Yes |

### 3.3. Hydrolytic Exoenzymes

A comparative analysis of the number of analyzed bacteria showed that lipolytic bacteria were dominant, with a higher average ($4.2 \times 10^2$ CFU mL$^{-1}$) than the other groups, havingthe highest abundance ($2.6 \times 10^3$ CFUmL$^{-1}$) (Table 4).

**Table 4.** Descriptive statistics of enzymatic groups (CFU mL$^{-1}$).

| Statistic | Amylolytic | Proteolytic | Lipolytic | Cellulolytic |
|---|---|---|---|---|
| Minimum | 0 | 0 | 0 | 0 |
| Maximum | 150 | 45 | 2650 | 120 |
| Mean | 19.913 | 11.588 | 426.888 | 18.55 |
| Variance ($n-1$) | 1286.345 | 184.922 | 259,498.814 | 759.715 |
| Standard deviation ($n$) | 35.414 | 13.428 | 503.002 | 27.216 |

The bacterial communities with lipolytic activity were better represented in the Colentina River (Figure 1).

According to the boxplot analysis (Figure 1), amylolyticbacterial communities were better represented in the Colentina River sector and especially in Lake Mogoșoaia. Also, the analysis highlighted the presence of peaks in Plumbuita Lakes (150 CFU mL$^{-1}$, in winter), Cernica (150 CFU mL$^{-1}$, in winter) and Fundeni (50 CFU mL$^{-1}$, in winter) which were higher compared to the other results of the study. They were also characterized by an increase in abundance throughout the study year, from an average of 3.25 CFU mL$^{-1}$ in spring to an average of 53.5 CFU mL$^{-1}$ in winter (Figure 2).

Proteolytic bacteria were less numerically represented compared to the other evaluated bacterial groups. The CFU values quantified during the study did not exceed a maximum of 45 CFU mL$^{-1}$ (Fundeni and Cernica lakes) and an average of 11.58 CFU mL$^{-1}$ (Table 4). From a spatial point of view, there is a slightly reduced number of proteolytic bacteria in the extreme stations of the river, Colentina River (4 CFU mL$^{-1}$) and Cernica (8.81 CFU mL$^{-1}$) (Figure 1). In the lakes belonging to the urban areas, the proteolytic bacteria registered a higher number, especially in the Fundeni (18.81 CFU mL$^{-1}$) and Mogoșoaia (13 CFU mL$^{-1}$) lakes, but the ecosystems did not differ significantly (ANOVA $F_{(5,34)}$ = 1.170; $p$ = 0.34).

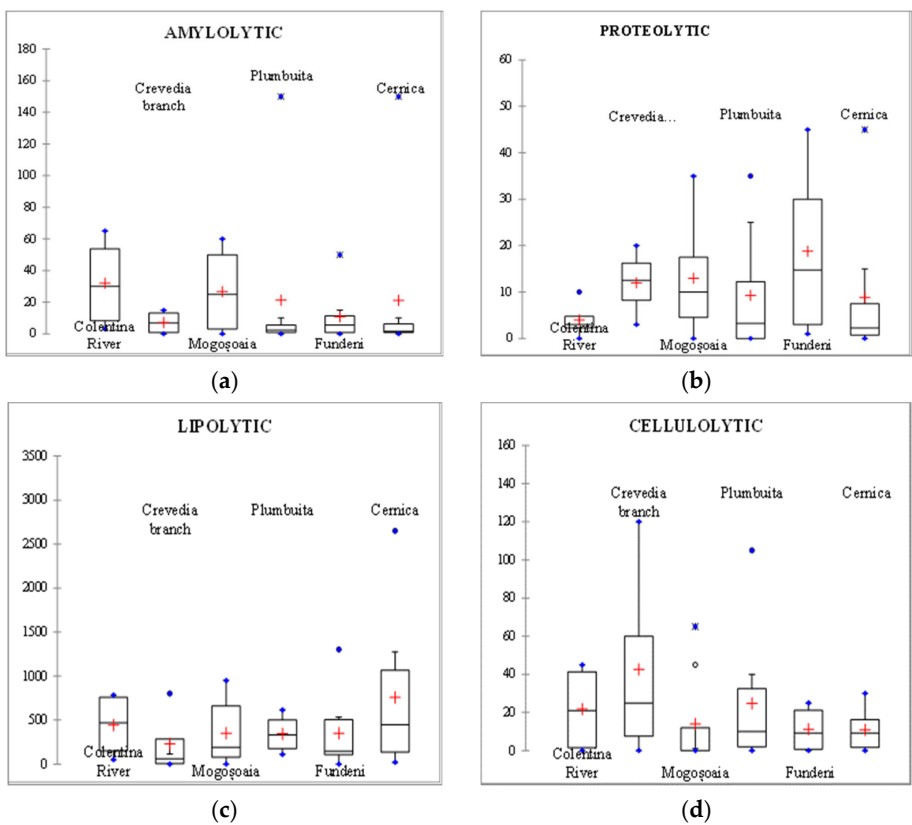

**Figure 1.** Amylolytic (**a**), proteolytic (**b**), lipolytic (**c**), and cellulolytic (**d**) bacteria (CFU mL−1). (Blue dots—outliers, red crosses—means, horizontal lines—quartiles 25%; 50%; 75%).

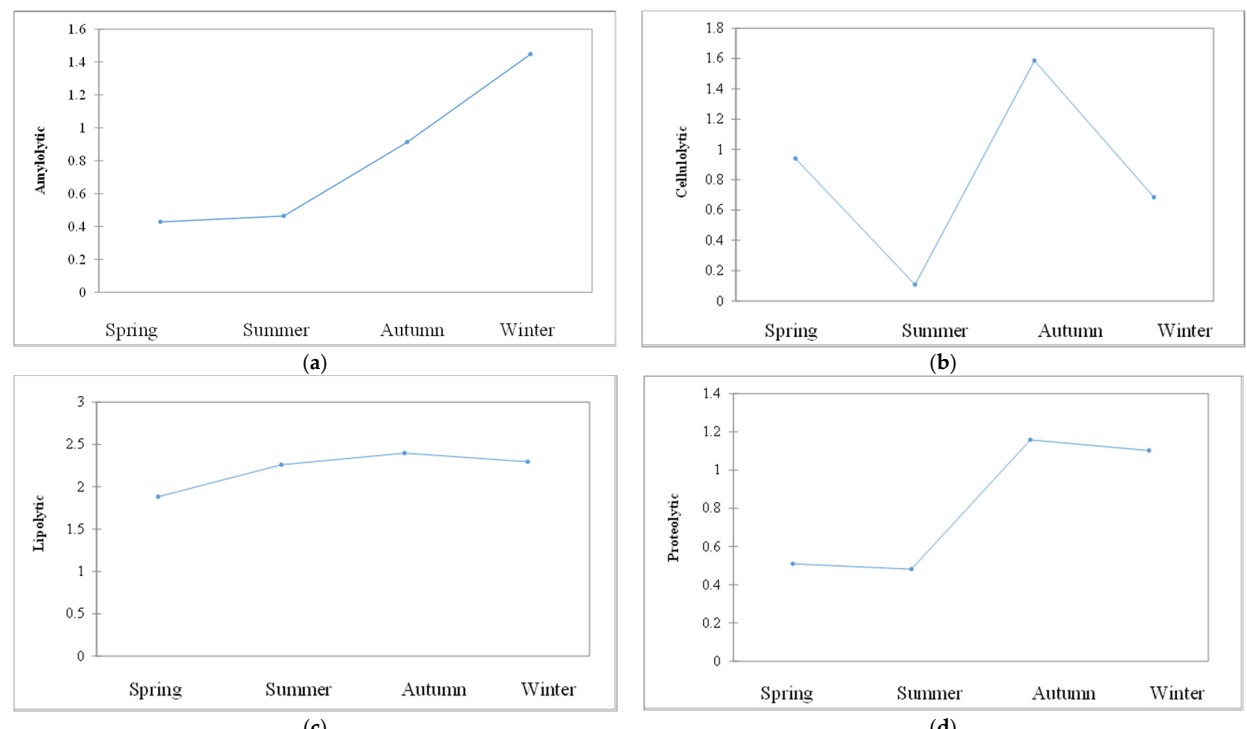

**Figure 2.** Seasonal means charts of amylolytic (**a**), cellulolytic (**b**), lipolytic (**c**), proteolytic (**d**) bacteria.

In the case of cellulolytic bacteria, the highest averages were found in the Crevedia branch (42.5 CFU mL$^{-1}$) and Plumbuita (24.81 CFU mL$^{-1}$). The maximum value was found in the Crevedia branch (120 CFU mL$^{-1}$) (Table 4), in spring.

The ANOVA test did not identify spatially significant differences in the four studied bacterial groups. In the case of temporal dynamics, there were differences for amylolytic ($F_{(3,36)}$ = 7.60, $p$ = 0.000), proteolytic ($F_{(3,36)}$ = 6.38, $p$ = 0.001) and cellulolytic ($F_{(3,36)}$ = 17.63, $p$ = 0.0001) bacteria (Figure 2).

Evidence for the response of the studied bacterial communities to the pressures of physico-chemical parameters on communities during the study period was provided by the multivariate RDA analysis. Thus, in Figure 3 it is shown that the physico-chemical factors introduced in the analysis explain in 85.14% proportion (canonical axis F1 68.45% + canonical axis F2 16.68%) the variation of the microbial community in the studied ecosystems. The Monte-Carlo permutation test allows us to accept the alternative hypothesis Ha with a significant relationship (XLSTAT pro, 2013. Data Analysis and Statistical Solution for Microsoft Excel. Paris, France: Addinsoft (soft))). Thus, the evaluated environmental parameters influenced the microbial activity. The first axis, F1, is related to the distribution of bacterial communities along a gradient of temperature, turbidity, depth and pH. Lipolytic, proteolytic, amylolytic and cellulolytic bacteria preferred deeper, warmer lakes, which had increased turbidity and higher pH (Figure 3, right).

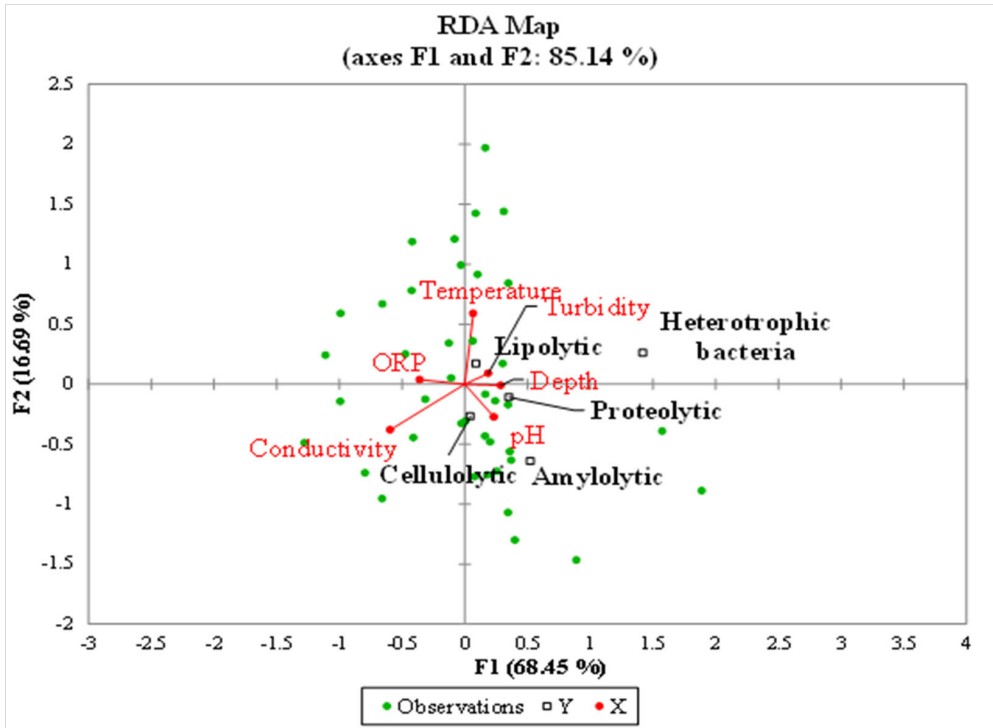

**Figure 3.** Redundancy analysis (RDA)triplot analysis of the relationships between microbial community parameters and measured environmental factors (legend observations = sampling stations, Y = response variables, X = explanatory variables).

The F2 axis revealed the preference of lipolytic bacteria for lakes with high temperatures, turbidity and high ORP. In contrast, proteolytic, cellulolytic and amylolytic bacteria develop better in waters with high conductivity and high pH.

## 4. Discussion

The chain of lakes of the Colentina River was generated from the blocking of the course of the river and rehabilitating its clogged riverbed. The resulting lake waterfall has a level difference of 49 m, keeping a slow flow direction, from upstream (station

SP0) to downstream (station SP9). Applying the ANOVA test, it was concluded that the ecosystems resulting from the dam show differences only in terms of depth, water velocity and DO content. These differences are explained by the fact that these lakes have a period of operation of decades, during which large amounts of alluvium carried by the waters that supply them have been deposited on the bottom of the lakes, or by rainfall from the lands bordering the shores. These alluvium deposits had the effect of decreasing the initial depth of the lakes, which facilitated the excessive growth of aquatic vegetation (where conditions favor its installation, in some lakes missing because the shores have anthropogenic interventions, such as concrete reinforcement). During the summer, episodes of algal bloom occur, which cause large variations of the oxygen concentration in the water. During the period when the lakes are shallow, from autumn (0.90 m) to spring (0.71 m), these deposits decompose under the action of microorganism communities.

More obvious are the temporal differences between ecosystems due to physico-chemical factors such as temperature, conductivity, salinity, TDS, ORP, pH and turbidity. These are particularly important factors in determining the seasonal dynamics of biological communities, especially microbial communities. The development of heterotrophic bacteria is dependent on both variations in temperature and pH [12], as well as factors that characterize water quality (conductivity, TDS, ORP, turbidity) [25]. Enzymatic activity measured in the sediment of Lake Erhai (China) showed significant positive correlations with the temperature measured in the water column [26]. Microbial metabolic rate increases with the increase of temperature, thus causing microorganisms to intensify the synthesis and secretion of enzymes [3].

According to Song [27], the number of heterotrophs decreased in spring when the ambient temperatures were below the optimum required, and subsequently, a slight tendency to increase the abundance of bacteria was observed. Under environmental conditions typical of temperate zones, the growth of bacteria is controlled by the thermic seasonal variations and by the concentration in the environment of metabolizable substrates.

Similarly, in the present study, the heterotrophic bacterial abundance due to the possible increased concentration of metabolizable substrate was observed during autumn or winter in all stations, except SP9. In this case, the degree of high heterotrophy was recorded in the summer.

The total count of heterotrophic bacteria in the water ranged from very low values of 25 $CFUmL^{-1}$ in Cernica Lake in spring to values of $2.3 \times 10^5$–$1.2 \times 10^6$ CFU $mL^{-1}$, recorded in Mogoșoaia Lake in autumn. The degradation processes of the aquatic vegetation intensify as the summer approaches its end and, together with the detritus resulting from the death of algal cells and zooplankton organisms, the accumulation of organic matter increases. An increasing trend of the number of heterotrophic bacteria can be observed, starting from July (values of $10^4$–$10^5$ CFU $mL^{-1}$ that are maintained during autumn and even increase in winter), which indicates the intensification of decomposition processes. The total count of heterotrophic bacteria in the water column found in a wetland did not exceed $2.3 \times 10^3$ CFU $mL^{-1}$ [28]. Data obtained in eutrophic and hypertrophic systems of the estuary lake type show that the greatest number of bacteria in water appeared in summer, reaching its minimum in winter [29]. These trends were statistically confirmed by the ANOVA test which revealed seasonal differences in the number of heterotrophic bacteria, as in the case of environmental factors. The higher values of CFU $mL^{-1}$ from stations SP1–SP9 can be explained by possible anthropogenic influences, such as direct discharges into the Colentina River by industrial units and the wastewater from population upstream of Bucharest. On the other hand, the bottoms of the lakes on the Colentina River have not been dredged for over 30 years, leading to the accumulation of large amounts of unhealthy mud.

In this research, the presence of bacteria potentially producing hydrolytic enzymes (lipases, proteases, amylases, cellulases) was studied. These enzymes are involved in cleaving polymers into monomeric units, much more accessible to bacterial metabolism, along with the return of biogenic elements to the upper trophic network [26,30].

In lakes belonging to urban areas, such as Fundeni and Mogoșoaia, proteolytic bacteria recorded a higher number. Actually, proteolytic, amylolytic and lipolytic bacteria play an important role in the decomposition of complex organic compounds resultingfrom household and industrial effluents.

In the case of temporal dynamics, amylolytic, proteolytic and cellulolytic bacteria showed differences. In contrast, other authors report that there is no regular pattern of seasonal distribution of the bacterial groups studied, except for proteolytic bacteria, which showed maximum density in autumn and minimum in winter. Seasonal variations are generally less marked for amylolytic and lipolytic groups [31]. Similarly, the results obtained in the present research revealed that lipolytic bacteria did not show differences in seasonal development, being dominant throughout the study period.

The physico-chemical factors are of great importance in shaping communities of microorganisms in lakes [32,33]. The influence of environmental factors on the synthesis and activity of enzymes produced by microorganisms is very complex and is usually not due to a single factor [3,34].

The RDA analysis showed that the depth, turbidity, conductivity and redox potential are associated with the F1 axis, while temperature and pH are associated with the F2 axis. A positive linear connection between lipolytic bacteria and temperature, turbidity and depth can be observed. Similarly, Souffreau et al. [32] found correlations of the physico-chemical factors (temperature, conductivity, pH, depth) with the bacterioplankton community in lakes located in various geographical regions. In our study, with an increase in temperature, the number of heterotrophs increased and to a lesser extent, also the lipolytic bacteria number. Cellulolytic and proteolytic bacteria have a directly proportional relationship with pH. The role of pH in structuring bacterial communities in aquatic ecosystems requires increased attention [35]. There was a linear relationship between conductivity and lipolytic bacteria, which means that increased conductivity values may contribute to the bacterial CFU number decrease. The same relationship can be observed in the case of the redox potential, which by increasing the valuescan decrease the CFU number of proteolytic, amylolytic and cellulolytic bacteria.

## 5. Conclusions

This study analyzed the metabolic potential of some functional groups of bacteria in aquatic urban systems. In order to achieve this goal, the evaluation of the abundance of communities of total heterotrophic bacteria in the water in relation to the physico-chemical factors was monitored. Seasonal differences were found, especially in the spring. The high values of the abundance of heterotrophs in the winter season reflect the urban activity in the respective stations. Screening for four types of extracellular hydrolytic enzymes with the potential for degradation of organic matter (amylases, lipases, proteases and cellulases) led to the conclusion that lipolytic bacteria were dominant in the studied ecosystems, while proteolytic bacteria were in a reduced number although present urbanized areas. The presence of cellulolytic bacteria is consistent with the presenceof macrophytic vegetation [36].

Regarding the physico-chemical factors that influence the bacterial communities, the results showed the significant role of temperature, conductivity, depth and pH on the variation in the abundance of the studied bacteria.

Due to the recognition of the increased role of the community of microorganisms in the functioning of ecosystems, it is necessary to continue our study to identify and quantify hydrolytic enzymes involved in the degradation of organic matter in urban lakes, and to test them in the degradation and elimination of hazardous substances. Anthropogenic ecosystems, degraded under various anthropogenic pressures, could select the most efficient variants of microbial strains capable of synthesizing enzymes for furtheruse in the bioremediation of waters.

**Author Contributions:** B.O. performs the experiments, collected literature data, sampling; R.C. (Rodica Catana) analysis of macrophytic vegetation, help for sampling; S.N., R.C. (Roxana Cojoc) perform the microbial experiments, collected literature data; A.I.L. performs the microbial experiments, correct the paper and graphic of paper, L.M. correct the statistical analysis and paper, L.F. perform statistical analysis, helps for sampling; R.R. performs for enzymatic activities, collected literature; M.E. design microbiological experiments and enzyme experiments, helps for sampling. M.M. writes the paper, helps for sampling. All authors have read and agreed to the published version of the manuscript.

**Funding:** This study was carried out in the framework of the projects no. RO1567-IBB02/2021 and no. RO1567-IBB05/2021 from the Institute of Biology Bucharest of Romanian Academy.

**Institutional Review Board Statement:** Not applicable.

**Informed Consent Statement:** Not applicable.

**Data Availability Statement:** The data and intellectual property belong to the journal.

**Acknowledgments:** The authors are grateful to the Institute of Biology Bucharest of Romanian Academy and ICUB—Faculty of Biology.

**Conflicts of Interest:** The author Bianca Ojovan is an employee of MDPI, however she does not work for the Journal Fermentation at the time of submission and publication.

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
