# Peer review of "Metabolic Potential of Some Functional Groups of Bacteria in Aquatic Urban Systems"

_fermentation, doi:10.3390/fermentation7040242_

Round 1

Reviewer 1 Report

General comments

The manuscript requires considerable language editing. Some sentences are confusing because the authors seem to use too many filling words that do not add much to the intended message. E.g., Line 64: “Most of them, urban aquatic ecosystems” could simply be “Most urban aquatic ecosystems”. Similarly, Line 69-70 “…leading to polluted waters, clogged, unsightly or even harmful to the human population.” does not make sense.

The abstract should be a standalone piece that gives the reader an overview of the work without details. In the present form, the abstract is insufficient. What was the rationale for this work or what was the aim? How were the results obtained? What is the implication or importance of the study? Any recommendation based on the results?

In the introductory section, the authors give a good background of existing literature but fail to identify the gaps. All these works have been done and so what? What are the authors intending to present which is not found in the literature? In other words, why are Lines 82-88 important? What new knowledge will it bring or what contribution will it make to existing knowledge on aquatic ecosystems?

Materials and methods:

A description of the sampling site is necessary. Why that specific site? What are the anthropogenic or natural activities around this site?

The sampling too was insufficient for this type of work, especially if anthropogenic activities are high around the sampling areas. Th authors collected a “single sample” per sampling site per season. This such sampling cannot be used as conclusive evidence on which recommendations can be made. This is because a single sampling could lead to a “hit or miss”, meaning that if something is seen, it does not necessarily mean that it would be there all the time. Similarly, not finding something in a single sample cannot exclude the potential presence of that thing in that environment. Replicate samples per site per season would have been the minimum.

It would also be important to know how far below the surface the water samples were collected.

Regarding the isolation of the bacteria, it would be good to mention what “agarized” media were used. Was this nutrient agar, heterotrophic plate count agar, etc.? In the papers cited (14 and 15) the authors targeted halophilic bacteria, meaning that their media may be different from the ones used in the current study. Please provide this information for reproducibility.

Section 2.3: This section is unnecessarily long. There is no need to explain what each type of analysis does (E.g., Line 145-148; check other instances). Simply indicate what test was used to analyse each set of data. You could supplement with references.

Results

In the Materials and Methods, the authors mention in Lines 97-98 that “Physico-chemical parameters were measured in situ. Temperature, salinity, pH and TDS were measured with a Hanna Instruments HI 9828 multi-parameter, while the depth and transparency of the lakes using the Secchi disk”

However, in the Results section, Lines 155-156, they say “(depth, water velocity and DO) and factors that varied seasonally (temperature, conductivity, salinity, TDS, ORP, pH and turbidity). How were the other parameters obtained?

Similarly, in Table 1, the authors have DO at the surface and DO at the bottom. However, this measurement was not specified in the materials and methods.

In the discussion section, the authors stress the role of environmental factors on the differences in bacterial abundance. However, their lakes are found in urban areas, suggesting that anthropogenic activities would highly influence nutrient loading and other parameters in the water, hence, influencing the microbial counts. As indicated in my previous comment on the description of the study site, it would be beneficial to indicate the anthropogenic activities around the sampling points to better understand these differences. These activities would also provide a clue to what type of organic matter may be loaded into the catchment.

Specific comments

Line 29: Delete “sequence of”

Line 30: Please change “circuit” to “cycling”

Line 38-39: “Several studies…functioning of the ecosystem.”. Please provide at least 3 or 4 references to back this claim, since you said, “several studies”.

Line 39-40: Please reference these roles.

Line 47-57: “Screening of microorganisms…[8,11]”. Please rephrase this section by changing the tense. It is presented as if it was work previously performed by the authors. If this is not the case, then I suggest rephrasing. If these studies were performed by the authors, then I suggest they clearly indicate this at the start of this paragraph. E.g., Line 48-49: …enzymes has been performed...has been analyzed.

Line 54: abundance and distribution…

Line 70-72 (In this frame comes…hydrolytic exoenzymes) and Line 72-75 (The role of …hydrolytic enzymes.” Say the same thing. Consider deleting one. The same applies to Line 75-76 and 80-81.

Line 84: I suggest changing “lakes arranged” to “human-made lakes”

Lines 160-171: Please present the results in a straightforward manner. Avoid unnecessary explanations (E.g., Lines 169-171).

Line 175-179: Please don’t explain the Tukey test.

The post hoc test table should be Table 3

Line 187: Which natural environments?

Lines 257-263: Repetition of the same thing. Please rephrase.

Author Response

Reviewer #1: The manuscript requires considerable language editing. Some sentences are confusing because the authors seem to use too many filling words that do not add much to the intended message. E.g., Line 64: “Most of them, urban aquatic ecosystems” could simply be “Most urban aquatic ecosystems”. Similarly, Line 69-70 “…leading to polluted waters, clogged, unsightly or even harmful to the human population.” does not make sense.

R: We would like to thank the reviewer for his or her appreciations on this work. We have addressed the answers to his or her questions below, in red, point by point.

 Agreed with the reviewer and corrected (see line 64).   In the line 69-70 we wrote clearer the phrase.

The language editing in the entire manuscript was improved.

The abstract should be a standalone piece that gives the reader an overview of the work without details. In the present form, the abstract is insufficient. What was the rationale for this work or what was the aim? How were the results obtained? What is the implication or importance of the study? Any recommendation based on the results?

R: We completed the abstract according to referee suggestions

In the introductory section, the authors give a good background of existing literature but fail to identify the gaps. All these works have been done and so what? What are the authors intending to present which is not found in the literature? In other words, why are Lines 82-88 important? What new knowledge will it bring or what contribution will it make to existing knowledge on aquatic ecosystems?

R Generally, urban ecosystems are a novelty in the field of evaluating their functioning in the ecological context known in the literature. This type of aquatic system, present in cities or peri urban areas, generates ecosystem services vital to human society. The degradation of the ecological state, the pollution with organic substances, even microplastics, leads to the alteration of the quality of the services offered by these ecosystems. The microbial component plays a particularly important role in the degradation of refractory substances. That is why finding suitable strains for bioremediation is welcome. Isolation of microbial strains from urban systems may provide a better solution to remedy them than strains obtained from other types of ecosystems that have not experienced major degradation.

Our results could serve as methods for bioremediation of urban ecosystems. Also, the awareness of the decision factors must be made on the basis of scientific research, the ecological methods being preferable to other chemical ones.

Materials and methods:

A description of the sampling site is necessary. Why that specific site? What are the anthropogenic or natural activities around this site?

R The Colentina River (101 km) crosses Ilfov County and Bucharest in the form of a chain of fifteen landscaped lakes connected by a system of hydrological control locks. 10 stations were established: a station in the unchanged area of ​​the river considered the reference area, 2 stations / lake in MogoÈ™oaia, Plumbuita, Fundeni lakes, a station in Cernica and a station at the exit of the arranged lakes system, in the river area. Also, in order to capture the input of a tributary of Colentina, a sampling station was established on the Crevedia arm (tributary of Colentina) (Cârstea, 2009; Stănescu & Gavriloaie, 2011). Thus, the result was a spatial sampling program in the direction of the rural-urban gradient, up to the downstream area of ​​the Colentina River.

A large part of the shores of these lakes are adjacent to homes and some are included in parks (Răducan and Streza, 2014). For this reason, the Colentina river has acquired over time an important role as an agreement area. Being accessible to the human population, the ecosystem services provided by the river are important. Among these we list the roles in agreement, fish exploitation, urban temperature regulation

The anthropogenic impact is diverse, coming mainly from inhabited areas and less from the agricultural area. It is known that urban ecosystems are subject to multiple pressures from anthropogenic activities and can reach these lakes in various ways, both through intentional discharges, rainfall or airway. (Completed in the manuscript at section 2.1 Description of sampling site).

Cârstea, M. (2009). Geographical description of the 2nd district of Bucharest. Present Environment and Sustainable development, 3. 243-249

Stănescu, S. V., È™i Gavriloaie, C. (2011). Aspecte privind vegetaÅ£ia ÅŸi fauna râului Colentina pe traseul din municipiul BucureÅŸti (România). Ecoterra, 27, 49-52

Răducan, V., și Streza, I. (2014). Militant ephemeral in Mogosoaia ensemble. Scientific Papers, LVIII, 359-368.

The sampling too was insufficient for this type of work, especially if anthropogenic activities are high around the sampling areas. Th authors collected a “single sample” per sampling site per season. This such sampling cannot be used as conclusive evidence on which recommendations can be made. This is because a single sampling could lead to a “hit or miss”, meaning that if something is seen, it does not necessarily mean that it would be there all the time. Similarly, not finding something in a single sample cannot exclude the potential presence of that thing in that environment. Replicate samples per site per season would have been the minimum.

R We agree with the comment of the referent that the sampling frequency was not extensive. Our study was intended to be a preliminary, a screening of microbial strains in urban lakes. The study will be extended in the next research program, in order to provide bioremediation recommendations to decision makers.

In our study we aimed to evaluate the features of heterotrophic communities with the identification of dominant groups under the pressures of environmental factors in urban ecosystems, selected in different periods and not a dynamic of their time. The variability of communities is very high both spatially and temporally, and of course a larger number of samples was valuable. However, we believe that it would not have significantly changed our results, in terms of the share of functional groups. See below the references about sampling program similar to ours.

 Kopylov, A.I., Kosolapov, D.B., Zabotkina, E.A. et al. Planktonic viruses, heterotrophic bacteria, and nanoflagellates in fresh and coastal marine waters of the Kara Sea Basin (the Arctic). Inland Water Biol 5, 241–249 (2012). https://doi.org/10.1134/S1995082912030054

Agedah EC, Ineyougha ER, Izah, SC, Orutugu, LA. Enumeration of total heterotrophic bacteria and some physico-chemical characteristics of surface water used for drinking sources in Wilberforce Island, Nigeria. Journal of Environmental Treatment Techniques, 2015; 3(1):28 - 34

It would also be important to know how far below the surface the water samples were collected.

R The samples were taken from the water column (see lines 104-105). The studied ecosystems are shallow lakes, with an average depth of 0.8 m (minimum of 0.2 and a maximum of 1.8 m). In order to better capture the microbial populations, the samples were collected on the entire water column.

Regarding the isolation of the bacteria, it would be good to mention what “agarized” media were used. Was this nutrient agar, heterotrophic plate count agar, etc.? In the papers cited (14 and 15) the authors targeted halophilic bacteria, meaning that their media may be different from the ones used in the current study. Please provide this information for reproducibility.

R MICRO

Section 2.3: This section is unnecessarily long. There is no need to explain what each type of analysis does (E.g., Line 145-148; check other instances). Simply indicate what test was used to analyse each set of data. You could supplement with references.

R We modified section 2.3 according to referee comment. We supplemented the references for RDA and ANOVA test. (See lines …..

Legendre, P., Oksanen, J. and ter Braak, C.J.F. (2011), Testing the significance of canonical axes in redundancy analysis. Methods in Ecology and Evolution, 2: 269-277. https://doi.org/10.1111/j.2041-210X.2010.00078.x

Capblancq T, Luu K, Blum MGB, Bazin E (2018) Evaluation of redundancy analysis to identify signatures of local adaptation. Molecular Ecology Resources, 18, 1223-1233

Quinn G.P., Keough M.J.,2002 Experimental Design and Data Analysis for Biologists, Cambridge University Press, 537 pp

Results

In the Materials and Methods, the authors mention in Lines 97-98 that “Physico-chemical parameters were measured in situ. Temperature, salinity, pH and TDS were measured with a Hanna Instruments HI 9828 multi-parameter, while the depth and transparency of the lakes using the Secchi disk”

However, in the Results section, Lines 155-156, they say “(depth, water velocity and DO) and factors that varied seasonally (temperature, conductivity, salinity, TDS, ORP, pH and turbidity). How were the other parameters obtained?

R We have provided below detailed information on how each environmental parameter was measured and how data collection was undertaken, as emphasised by the reviewer. We can only hope that the latest version presents these aspects clearer.

Physico-chemical parameters were measured in situ. Temperature, salinity, pH and TDS, DO were measured with a Hanna Instruments HI 9828 multi-parameter, while the depth and transparency of the lakes using the Secchi disk. Also, the water velocity was determined using a basic flowmeter, Geopacks Flowmeter (MFP51).

Similarly, in Table 1, the authors have DO at the surface and DO at the bottom. However, this measurement was not specified in the materials and methods.

R The oxygen concentration DO at the surface and at the bottom were measured in situ, using a Hanna Instruments HI 9828 multi-parameter (Lines 102-107).

In the discussion section, the authors stress the role of environmental factors on the differences in bacterial abundance. However, their lakes are found in urban areas, suggesting that anthropogenic activities would highly influence nutrient loading and other parameters in the water, hence, influencing the microbial counts. As indicated in my previous comment on the description of the study site, it would be beneficial to indicate the anthropogenic activities around the sampling points to better understand these differences. These activities would also provide a clue to what type of organic matter may be loaded into the catchment.

R: All the recommendation have been included accordingly.

Specific comments

R We modified in the manuscript all the specific comments.

Reviewer 2 Report

This paper deals with the distribution and metabolic characteristics of heterotrophic bacteria in an urban system (the Colentina River) impacted by anthropic activity along its course, through a seasonal survey. The specific aims were to find the physico-chemical parameters, affecting the dynamics of heterotrophic bacteria, and to detect the ability of bacterial isolates to degrade organic compounds through their metabolic profiles (proteolytic, lipolytic, amilolytic, and cellulolytic activities) with the perspective of their use for bioremediation of polluted waters.

The subject of this study is not particularly original, but it could be of interest for the potential application of the obtained results.

Minor revisions are needed, regarding the English language and the Figures

Abstract

line 14, in the water; remove "in the field"

line 16, in the winter season are related to the human activity at the sampling stations

line 19, while proteolytic bacteria were observed in low numbers

Introduction

line 29, the process of organic matter decomposition

line 30, microorganisms in the organic matter pathways along the food web

line 32, in recent decades they have been recognized

line 41, insert one reference

line 44, needed to higher trophic levels

line 50, The main aim of these studies

line 52, Bacterial isolates, having amylase, lipase and cellulolytic activity. were obtained

lines 57-58, microbial equipment of enzymes

lines 61-62, please explain better the expression on the banks that can degrade and reach the water mass

lines 63-64, Generally, urban ecosystems are a novel research fields because their ecological functioning is not fully known.

line 69, exceeds the buffering capacity

lines 70-71, In this frame fits the potential bioremediation ability of the microbial community  through its ability

lines 73-74, organic matter into simpler inorganic compounds within biogeochemical cycles is related to the

lines 80-81 repeat the same sentence of lines 75-76 (Depending on the compounds...)

line 77, constitutive enzymes, that are synthesized whether or not their specific substrates are present; other enzymes, inducible ones, are produced only when their substrate is available

Line 82, The present study (instead of paper)

line 84, This investigation had two specific objectives, the first addressed the abundance of total heterotrophic bacteria in water in relation to the physico-chemical factors; the second one focused on the screening for extracellular hydrolytic enzymes

Materials and methods

Line 91 have been set up along the Colentina River

line 92,The numbering of sampling stations was assigned from upstream

line 96, November to cover the seasonal periods

line 98, Total Dissolved Solids (TDS), first in full, then the acronym

Oxidation-reduction potential (ORP) must be reported here, with the method used for its measurement

Line 108, it sounds me very strange that the activities of bacteria isolated from aquatic compartments are measured at 37°C that is a temperature more suitable for the detection of pathogens, instead of environmental autoctonous microflora (growing at 22-24°C), please check this point

line 114, increased both on the surface and in depth? please explain better, does this refer on the culture plate? (if the spread method was used, the isolated were grown on the surface only, if you used the pour plate method, bacteria were grown also within the culture media)

line 119, Descriptive statistical analysis and box-plots were performed

line 121, highlight the main results

lines 123-124, on the distribution of microbial communities was assessed with a multivariate analysis.

lines 126-127, multicollinearity matrix was generated.

line 144, the analysis has a significant

line 145, The principle of the analysis of variance

Results

line 155 among ecosystems

line 164, the second value in terms of magnitude order is recorded

line 179, at stations

Post hoc Tukey- this is Table 3 instead of table 1

lines 193- 196, move these sentences after Table 4, they refer to Fig. 2 box-plots and explain better the sentence (A better representation of the bacterial communities we find...)

line 196, spring to winter (from 219 to 580 CFU mL-1 respectively)

line 211, but the ecosystems did not differ significantly is in contrast with the presence of asterisks in the figure, please check

line 217. differences for amylolytic

Figure 2 has too small size characters in both x and y axes

lines 224-225, The response of the....parameters was highlighted

line 226. it is shown that

line 230. Ha, with a significant relationship

Figure 3, pH label is not clearly visible

Discussion

line 242, River was generated from the blocking of the course

line 246, DO instead of dissolved oxygen

line 255, when the lakes are low (is the right meaning: have low water levels?)

line 270, conditions typical of temperate zones, ... is controlled by the thermic seasonal variations

line 272, Similarly, in the present study,  the heterotrophic bacterial abundance ...

line 288, ANOVA test which reveal seasonal difference

line 314, to these references, please add other ones regarding the effects of environmental parameters on microbial community abundance and activity of lacustrine environments, i.e.

Caruso G., Monticelli L., Azzaro F., Azzaro M., Decembrini F., La Ferla R., Leonardi M., Zaccone R. (2005) Dynamics of extracellular enzymatic activities in a shallow Mediterranean ecosystem (Tindari ponds, Sicily). Marine and Freshwater Research 56, 173-88.https://doi.org/10.1071/MF04049;

Zaccone R., M. Azzaro, F. Azzaro, A. Bergamasco, G. Caruso, M. Leonardi, R. La Ferla, G. Maimone, M. Mancuso, L. S. Monticelli, F. Raffa, E. Crisafi2014 Seasonal Dynamics of Prokaryotic Abundance and Activities in Relation to Environmental Parameters in a Transitional Aquatic Ecosystem (Cape Peloro, Italy)- Microbial Ecology, 67:45–56 DOI 10.1007/s00248-013-0307-z

Caruso G., Azzaro M., Monticelli L.S., Leonardi M., Cao C., Zhou Y., Song, C. 2017, Seasonal variations in microbial parameters and trophic states in a large Chinese shallow lake (Lake Taihu). Fresenius Environmental Bulletin.26 No.1-A pp.785-796 

Conclusions, line 334, This study

line 337, factors was monitored

line 344, bacteria is consistent with the presence of macrophyte vegetation

References

Ref. 11 is Cunha A (not Chunta)

Author Response

Reviewer #2:

R: We would like to thank the reviewer for his or her appreciations on this work. We have addressed the answers to his or her questions.  All the recommendation have been included accordingly.

Specific comments

R: We modified in the manuscript all the specific comments.

Round 2

Reviewer 1 Report

The authors have failed to work on the language of the manuscript as suggested previously. The manuscript skill requires extensive language editing. Just look at Lines 131-134.

Most importantly, several comments raised have not been addressed, although the authors indicate that they have done so. For example, the comments raised on the discussion regarding anthropogenic activities.

Abstract

Here, the authors still fail to present the abstract in a proper manner. The additions they have made contains numerous language errors. I strongly advise that the authors read the “Instruction for Authors” which states that:

Abstract: A single paragraph of about 200 words maximum. For research articles, abstracts should give a pertinent overview of the work. We strongly encourage authors to use the following style of structured abstracts, but without headings: (1) Background: Place the question addressed in a broad context and highlight the purpose of the study; (2) Methods: briefly describe the main methods or treatments applied; (3) Results: summarize the article's main findings; (4) Conclusions: indicate the main conclusions or interpretations. The abstract should be an objective representation of the article and it must not contain results that are not presented and substantiated in the main text and should not exaggerate the main conclusions.”

Author Response

All required changes have been made.